# The Method of Direct and Reverse Phase Portraits as a Tool for Systematizing the Results of Studies of Phase Transitions in Solutions of Thermosensitive Polymers

**DOI:** 10.3390/gels10060395

**Published:** 2024-06-11

**Authors:** Akhat Bakirov, Eldar Kopishev, Kaisarali Kadyrzhan, Elvira Donbaeva, Aigerim Zhaxybayeva, Marat Duisembiyev, Faiziya Suyundikova, Ibragim Suleimenov

**Affiliations:** 1Department of Chemistry and Technology of Organic Substances, Natural Compounds and Polymers, Faculty of Chemistry and Chemical Technology, Al Farabi Kazakh National University, Almaty 050040, Kazakhstan; a.bakirov@aues.kz; 2Department of Telecommunication Engineering, Institute of Communications and Space Engineering, Gumarbek Daukeev Almaty University of Power Engineering and Communications, Almaty 050040, Kazakhstan; kaisarali1997ss@gmail.com; 3Department of Chemistry, Faculty of Natural Sciences, L.N. Gumilyov Eurasian National University, Astana 010000, Kazakhstan; donbayeva_ek@enu.kz (E.D.); zhaxybayeva_ag_1@enu.kz (A.Z.); duisembiyev_mzh_1@enu.kz (M.D.); suyundikova_fo@enu.kz (F.S.); 4National Engineering Academy of the Republic of Kazakhstan, Almaty 050010, Kazakhstan

**Keywords:** thermosensitive polymers, phase transitions, inverse phase portrait method, data systematization, classification accuracy, turibidimetry, thermograms

## Abstract

It is shown that a more than significant amount of experimental data obtained in the field of studying systems based on thermosensitive hydrophilic polymers and reflected in the literature over the past decades makes the issue of their systematization and classification relevant. This, in turn, makes relevant the question of choosing the appropriate classification criteria. It is shown that the basic classification feature can be the number of phase transition stages, which can vary from one to four or more depending on the nature of the temperature-sensitive system. In this work, the method of inverse phase portraits is proposed for the first time. It was intended, among other things, to identify the number of phase transition stages. Moreover, the accuracy of this method significantly exceeds the accuracy of the previously used method of direct phase portraits since, for the first time, the operation of numerical differentiation is replaced by the operation of numerical integration. A specific example of the application of the proposed method for the analysis of a previously studied temperature-sensitive system is presented. It is shown that this method also allows for a quantitative comparison between the results obtained by the differential calorimetry method and the turbidimetry method. Issues related to increasing the resolution of the method of direct phase portraits are discussed.

## 1. Introduction

Phase transitions in solutions of thermosensitive polymers have attracted steady interest from researchers over the past few decades, so a query in the WoS database using the keywords “thermosensitive”, “thermoresponsive polymer” yields 18,886 results; their distribution by year is shown in Figure 1. Polymer-based systems in which such transitions occur are of interest for many practical applications. Thus, applications in oil and gas production are considered in the literature [1], in electronics [2], as well as in the energy field [3]. Traditional interest in the use of these systems in bioengineering and biomedical applications remains [4,5,6,7,8,9]. Heat-sensitive polymers are also of interest for environmental purposes; in particular, they can significantly improve [10] the continuous cycle of producing pure water using compression and swelling of hydrogels [11], provide energy savings for smart houses [12,13], etc.

To date, very extensive experimental material has been accumulated, reflecting the nature of phase transitions in systems based on thermosensitive polymers (their solutions as well as sols and gels), which is discussed in detail below. However, the range of experimental techniques used to study them remains quite limited. As a rule, methods such as viscometry, differential scanning calorimetry (DSC), and spectroscopy are used [14,15,16], as well as turibidimetry.

Thermosensitive polymers, as a rule, are synthesized on the basis of a fairly narrow range of starting substances, including those used in combination with other high-molecular and/or low-molecular components or combinations thereof. This follows from the review of articles presented below for the last five years, in which the classification is carried out according to the composition of the main component.

Methylcellulose (MC) is a cellulose derivative that has been extensively investigated for biomedical applications [14,17,18,19] and even for automatic fire extinguishing systems with controlled water release [20,21]. Stabenfeldt et al. [22,23] functionalized methylcellulose with the protein laminin, aiming towards the creation of a bioactive scaffold for neural tissue engineering. They reported that the laminin-functionalized oxidized methylcellulose (OXMC + LN) hydrogel promoted neuronal cell adhesion and showed higher cell viability rates than MC, oxidized methylcellulose (OXMC), or laminin-functionalized methylcellulose (MC + LN). The optimal ratio for hydrogel formation was determined by mixing agarose-carboxymethyl cellulose with gelatin at concentrations of 1% *w*/*v*, 3% *w*/*v*, and 5% *w*/*v*. It was observed that blends (C2-A0.5-G1 and C2-A1-G1), containing 2% *w*/*v* carboxymethyl cellulose, 0.5% or 1% *w*/*v* agarose, and 1% *w*/*v* gelatin, produced superior hydrogels with enhanced stability for up to 21 days in DPBS at 37 °C. These bioinks were successfully used in extrusion bioprinting, demonstrating the ability to print intricate 3D patterns [24].

Chitosan: Temperature-sensitive hydrogels such as hydroxybutyl chitosan gel (HBC) cross-linked with graphene oxide (GO) are promising for Pingyangmycin (PYM) rectal applications. HBC/GO@PYM exhibits 8 times higher drug loading than HBC@PYM [25]. An innovative thermosensitive injectable chitosan/poloxamer-based hydrogel has been developed for cardiac tissue regeneration. Gold nanoparticles (AuNPs) interacting with oxidized bacterial nanocellulose fibers (OBCs) were incorporated into the hydrogel at a percentage of 1% to modify its mechanical and electrical properties. In vitro biocompatibility testing using H9C2 cells showed high cell viability (>90% after 72 h) and efficient adhesion [26]. Mohamed et al. prepared complex polyelectrolyte chitosan/pectin nanoparticles by ionic gelation, coated them with phospholipid, and then incorporated them into an optimized thermosensitive injectable gel to improve the bioavailability of terbutaline sulphate [27]. The study by Stanzione et al. proposed a chitosan-based system that gels with the addition of beta-glycerophosphate and sodium bicarbonate solutions and is suitable for cell encapsulation. The systems gel under physiological conditions (37 degrees Celsius, pH 7.4), are stable in vitro for up to three weeks, have high swelling coefficients, and have mechanical stiffness suitable for cell encapsulation [28]. In this study, thermosensitive injectable hydrogels based on chitosan and polygalacturonic acid (PgA) were prepared and characterized. The biocompatibility and gelation time of the hydrogel were optimized by the addition of beta-glycerophosphate (beta GP) and hydroxyapatite. The results showed that hydrothermal treatment and incorporation of gelatin into the chitosan-beta GP hydrogel system improved the bioactivity and mechanical properties of the hydrogel [29]. A chitosan-gelatin-based thermosensitive hydrogel containing 5FU-alginate nanoparticles (5FU) has been formulated for the effective and sustained delivery of 5FU to the skin. Chitosan-gelatin-based hydrogels containing 5FU have exceptional properties and can be used for the sustained delivery of 5FU for the treatment of local skin cancers [30]. Highly deacetylated chitosan and N-isopropylacrylamide (NIPAM) were loaded with RALA/pEGFP-N1 nanoparticles, and release was studied. The nucleic acid cargo remained functional, as confirmed by the successful transfection of the NCTC-929 fibroblast cell line [31]. Another study aimed to develop and compare the performance of different chitosan hydrogel-based bioinks for 3D bioprinting. It was found that the type of gelling agent had a greater influence on these properties than the type of solvent [32].

Dextran: A new water-soluble thermosensitive star-like copolymer, dextran-graft-poly-N-isopropylacrylamide (Dg-PNIPAM), was created and characterized by various techniques (size-exclusion chromatography, differential scanning calorimetry, Fourier-transform infrared (FTIR) spectroscopy, dynamic light scattering (DLS) spectroscopy, and confocal laser scanning microscopy) [33].

Xyloglucan: In a series of papers, Dalvi et al. created Rufinamide nanocrystals and suspended them in a heat-sensitive xyloglucan-based gel to improve distribution from the nose to the brain [34,35,36]. In this work [37], an in situ gelling system for dual-response triamcinolone acetonide eye drops was developed and optimized using reacted tamarind seed xyloglucan and kappa-carrageenan polymers. The aim of the research was to develop a mucoadhesive gel formulation of sumatriptan for rectal administration using a combination of mucoadhesive polymers, poloxamer 407, and poloxamer 188. This formulation is designed to prolong the duration of drug residence in the rectum, thereby improving efficacy in the treatment of migraine [38]. Another paper reports the development of bioinspired thermosetting hydrogels composed of xyloglucan and cellulose nanocrystals for use as wound healing dressings, drug delivery vehicles, implants, etc. [39].

Gelatin is one of the most important materials on the basis of which thermosensitive polymers are produced. Ghanbari et al. developed hybrid injectable and biodegradable hydrogels based on oxidized alginate/gelatin with the addition of nitrogen-doped carbon dots as reinforcement and crosslinked with carbodiimide/N-hydroxysuccinimide. Graphene quantum dots significantly influence the degradation and interaction with cells and hydrogels [40,41,42].

In a study aimed at developing thermosensitive nanocomposite hydrogels based on a gelatin matrix with acetaminophen-loaded nanoparticles of the thermosensitive polymer PNIPAM. The system exhibited a reversible solution-to-gel transition around 37 degrees Celsius. The hydrogels containing a low ratio of PNIPAM to gelatin showed a more uniform and delayed release, indicating their good potential for drug delivery purposes [43,44].

An injectable self-healing hydrogel based on two extracellular matrix-derived biopolymers, gelatin and chondroitin sulphate, is being developed for use as a surgical adhesive to seal or reconnect torn tissue [45].

Vasya et al. developed microcarriers composed of near-infrared (NIR) light-absorbing GO, thermosensitive PNIPAM and biocompatible gelatin methacrylate. Under NIR light, the microcarriers have great potential for clinical applications in tissue repair [46,47].

Multifunctional temperature-sensitive microcarriers made from PNIPAM, poly(ethylene glycol) diacrylate, and methacryloyl gelatin for efficient cell expansion for industrial-scale cultivation of therapeutic cells [48].

Static substrates to simulate the dynamic behavior of the myocardium were cultured in vitro with The hydrogels were formed by crosslinking thermoresponsive poly(N-isopropylacrylamide-s-2-hydroxypropyl methacrylate-s-mercaptoethyl acrylate) synthesized using reversible addition-fragmentation chain transfer polymerization with gelatin [49].

The aim of another study was to develop thermoresponsive hydrogels made of poloxamer (P407) and polyvinyl alcohol for the purpose of delivering mupirocin nanoparticles to aid in wound healing. The mupirocin nanoparticles, which contained both the drug and either gelatin or poly(acrylic acid), were prepared through the process of spray drying [50].

This study presents a new hydrogel made up of bacterial cellulose, gelatin, and acid-treated halloysite nanotubes (Hal). The hydrogel aims to serve a dual purpose of providing structural support for tissue regeneration and enabling localized drug delivery [51].

This study employed threefold dendritic oligoethylene glycols (OEGs) to modify gelatin, a biomacromolecule known for its biocompatibility and bioactivity. The modification resulted in the creation of dendronized gelatins (GelG1s) with radial amphiphilic characteristics. The GelG1s exhibit either lower critical solution temperature (LCST) or upper critical solution temperature (UCST) thermoresponsive behavior, depending on the grafting ratio of the dendritic OEGs [52].

Chemically cross-linked gelatin microgels (ranging from 5 to 15 μm) were created by combining NIPAM, methylallyl polyethylene glycol, and GO via emulsion. In situ free radical polymerization resulted in a physical network within the microgels, forming double network (DN) microgels. These DN microgels exhibit thermosensitive properties and are responsive to NIR stimuli. In vitro studies have demonstrated that NIR irradiation efficiently induced the release of ropivacaine from the DN microgels, suggesting their potential in pain management [53].

### 1.1. Poly(ethylene glycol)—PEG

In the series of works conducted by Mueller-Buschbaum et al. [54,55,56,57] the kinetic rehydration of thin di-block copolymer poly(diethylene glycol monomethyl ether methacrylate)-block-polyPEG methyl ether methacrylate (PO2-b-PO300) films containing two thermoresponsive components is studied by in situ neutron reflectivity with different thermal stimuli in the D2O vapor atmosphere.

Sobczak et al.’s study on the development of injectable, thermosensitive hydrogel drug delivery systems for the release of 5-fluorouracil and paclitaxel. Biodegradable triblock copolymers were synthesized via ring-opening polymerization of epsilon-caprolactone or rac-lactide with PEG and zirconium(IV) acetylacetonate (Zr(acac)4) as co-initiator and catalyst [58,59,60].

Suljovrujic et al. have developed thermoresponsive hydrogels for biomedical applications. The hydrogels were created using oligo(ethylene glycol) dimethacrylate crosslinking monomers, and the study explored varying monomer/solvent ratios (10 to 100 wt%) [61,62,63].

Teodorescu et al. investigate the control of thermal response in aqueous solutions of PNIPAM-PEG-PNIPAM triblock copolymers through factors such as polymer concentration, composition, and additives. The study compares the effects of additives on Pluronic L61 and PNIPAM copolymers, revealing similar trends except for certain alcohols due to differences in polymer main chain structure [64,65,66,67].

A group of researchers have developed a sprayable hydrogel formulation that undergoes in situ gelation upon exposure to skin temperature. This is designed for effective wound dressing. The hydrogel consists of a copolymer blend, namely (PNIPAM (166)-co-n-butyl acrylate(9))-PEG-(PNIPAM (166)-co-n-butyl acrylate(9)), and includes silver-nanoparticle-decorated reduced GO nanosheets [68].

A thermoresponsive pseudo-block copolymer was synthesized by creating a PNIPAM star polymer with a beta-cyclodextrin core and an adamantyl-terminated PEG polymer. The polymer self-assembled through host-guest complexation and underwent the temperature-induced formation of supramolecular micelles [69].

There is a significant amount of research dedicated to developing drug formulations using triblock copolymers of polylactide and PEG [70,71,72,73,74].

### 1.2. PEO/PPO-Based Systems, Poly(ethylene oxide)—b-poly(propylene oxide)-b-poly(ethylene oxide) (PEO-PPO-PEO) Also Known as Poloxamer, Pluronic, Kolliphor, and Synperonic

Kozicki et al. have published a series of papers on the development and optimization of a radiochromic gel dosimeter using Pluronic F127 (F127). The potassium iodide based dosimeter with the F127 matrix shows potential for three-dimensional radiotherapy dosimetry. Another innovation is the printing of radiochromic hydrogels on woven fabrics for UV sensitivity using nitroblue tetrazolium chloride in the F127 matrix [75,76,77,78,79,80,81,82,83,84].

Alexandridis et al. investigated the properties of F127. Their study examines the complex interactions between surfactants and amphiphilic polymers, focusing on sodium dodecyl sulphate (SDS), F127, and Pluronic P123. In particular, F127 shows an earlier interaction with SDS than P123 due to differences in hydrophobicity. These results provide information for their applications in areas such as energy storage and solvent development [85,86,87,88,89].

Maintaining the integrity of cell membranes is critical for cell survival, and poloxamer 188 (P188) has been shown to be effective in protecting cell membranes from various insults. The results showed that P188, together with the PEO homopolymer (PEO8.4K), penetrates the bilayer without compromising its integrity, with the PEO providing the binding to the membranes. Collectively, these results shed light on the mechanisms underlying the membrane interactions and protective effects of P188 and related polymers, informing their potential therapeutic applications for membrane stabilization and drug delivery [90,91,92,93].

Aswal et al. studied F127 modified by the addition of PNIPAM at both ends of the PEO, resulting in the formation of a pentablock copolymer (PN41). The study of PN41 adsorption on hydrophobic gold surfaces revealed a temperature-dependent behavior influenced by polymer-solvent interactions. Pluronics F38, F68, F88, F98, and F108 with 80% hydrophilicity formed micelles with the PO core and hydrated EO corona at ambient temperature. The studies shed light on the behavior of PEO-PPO-PEO copolymers under temperature and salt and their applications in drug delivery and energy storage devices [94,95,96,97,98].

Poloxamer P407 has also been sufficiently investigated. There are studies on the development of plant-based larvicidal formulations with P407 against mosquitoes [99]. P407 is used to synthesize amphiphilic polyetherurethanes (PEU) for biomedical applications [100,101]. Poloxamer modified with dihydrocaffeic acid (P-DA) was developed to demonstrate improved mechanical strength, erosion resistance, tissue adhesion, and stability [102]. In addition, cannabidiol nanomicelles produced using P407 demonstrated anti-inflammatory effects in cell and animal studies, with greater efficacy and safety compared to CBD alone [103]. The reverse poloxamer (RP407) showed improved gelation properties compared to P407, which is relevant for applications such as drug delivery and protein stabilization. Mixing P407 and RP407 altered the size and packing of micelles, providing customized gelation behavior for various biomedical applications [104].

### 1.3. Poly(vinylcaprolactam)—PVCL

Venkatesu et al. studied the interaction of PVCL, their interactions with additives such as 1-allyl-3-methylimidazolium bromide, multi-walled carbon nanotubes, methylamine-based osmolytes, PNIPAM, deep eutectic solvents, and tryptophan-based amino acid ionic liquids, and their potential biomedical applications, laying the groundwork for innovative advances in drug delivery and biotechnology [105,106,107,108,109,110,111,112,113,114].

A new synthesis was tested by Pich et al., which yielded stimuli-responsive PVCL microgels decorated with zwitterionic poly(sulfobetaine), poly(phosphobetaine), poly(carboxybetaine), or selenium-functionalized chains via reversible addition-fragmentation chain transfer (RAFT) precipitation polymerization [115,116,117,118,119,120].

Coclite et al. illustrate the versatile synthesis and tunability of thermoresponsive polymers based on PVCL coupled to di(ethylene glycol)divinyl ether, Eudragit, or laser-induced graphene by initiating chemical vapor deposition, offering control over properties for drug delivery, sensors, actuators, and energy applications [121,122,123,124].

Research by Licea-Claverie et al. investigates the preparation of PVCL-based nanogels modified with galactose and loaded with cisplatin or doxorubicin (DOX) together with gold nanorods (GNRDs) using radical photopolymerization. Nanoparticle loads of up to 1 wt% were achieved in the study. In addition, thermoresponsive diblock copolymers consisting of PEG and PVCL blocks were prepared by RAFT polymerization. These copolymers effectively stabilized GNRDs and exhibited temperature-sensitive aggregation tendencies, indicating promising prospects in drug delivery systems for photothermal therapy [125,126,127,128].

To understand the physicochemical properties of temperature-responsive systems, Sala et al. prepared nanocomposites consisting of PVCL and silica nanoparticles of different sizes and concentrations. In addition, we developed thermosensitive nanocomposites using PVCL and silica particles that showed minimal toxicity and potential for integration with bone tissue, as well as serving as effective drug delivery systems for hydrophobic and DOX drugs. These nanocomposites exhibited controlled release mechanisms and showed encouraging anti-cancer effects in vitro [129,130,131,132].

Soft PVCL nanogels decorated with AuNPs or encapsulated GNRDs were investigated as carriers for small molecules and DOX. Nanoparticles were prepared for controlled drug delivery, with particle size and thermoresponsive behavior dependent on copolymer length and concentration. These results demonstrate the feasibility of using PVCL microgels as templates and stabilizers at high temperatures and offer novel synthesis strategies for hybrid microgels applicable in various fields of nanotechnology, including catalysis, sensing, and therapy [133,134,135,136,137].

It can be seen that, despite the limited range of basic thermosensitive components, the substances described in the literature are very diverse. Such a large amount of experimental material, as reflected in scattered articles, already requires systematization. It is also important that, to study phase transitions in systems of the type under consideration, different authors use different groups of experimental techniques.

Consequently, the question arises about the classification criteria on the basis of which such systematization can be carried out, as well as about the tools that would allow determining these criteria. Considering, however, that systems undergoing a phase transition with temperature variations are very diverse, at this stage of research, it is inappropriate to raise the question of quantitative classification criteria.

It seems much more productive to find criteria that ensure the formation of a qualitative classification. We also emphasize that, as follows from the above literature review, phase transitions can occur through different mechanisms; moreover, different stages of a phase transition can also occur through different mechanisms (in the case when it occurs in several stages). Consequently, it seems premature to raise the question of quantitative classification criteria. Qualitative classification criteria are also applicable in the cases where mechanisms causing phase transitions are of a different nature.

This work shows that one of the most significant qualitative criteria allowing for the classification of systems based on hydrophilic polymers is the number of stages inherent in the phase transition. To substantiate this conclusion, this work also developed a method of inverse phase portraits, which is used along with the method of direct phase portraits used in the literature.

We also emphasize that the method of inverse phase portraits, which allows for a convenient and visual comparison of the results of calorimetric and turbidimetric measurements, is proposed in this work for the first time. The very concept of “reverse phase portrait” is also used for the first time.

## 2. Results and Discussion

### 2.1. Results

#### 2.1.1. Rationale for Using the Inverse Phase Portrait Method to Study Phase Transitions in Solutions of Thermosensitive Polymers

Figure 2 shows thermograms of aqueous solutions of systems based on thermosensitive polymers obtained in [138]. Figure 3 shows the transparency dependences of the same solutions obtained in the cited work.

The type of curves presented in Figure 2 qualitatively coincide with the types of curves that can be obtained by numerical differentiation of the curves shown in Figure 3.

This conclusion directly confirms the following construction. Let us move from the dependence $C_{p}(T)$, where $C_{p}$ is the heat flux directly recorded by differential scanning calorimetry, to the dependence S(T), where $S\left(T\right)=\int_{T_{m}}^{T}C_{p}dT$.

From obvious physical considerations (phase transitions in a system of arbitrary nature are associated with energy costs for changing the state of the system elements), the function S(T) can be compared with any other physical quantity that describes the resulting change in the state of the system. In relation to the case under consideration, it is permissible to use as such a function the function D(T)—the dependence of the turbidity of the solution on temperature. Indeed, the turbidity of the solution in question (provided that the phase transition leads to an increase in the proportion of insoluble components in the system) depends on the cumulative changes that occur in the system with increasing temperature.

The function D(T), under the assumption that the phase transition caused by an increase in temperature is characterized by a transition from an almost transparent solution to an almost opaque one, can be reduced to a transmittance function $T_{t}(T)$ in accordance with the following formula:
(1)$$T_{t}\left(T\right)=\left(1-\frac{D\left(T\right)}{D_{0}}\right)\cdot 100\%$$
where $D_{0}$ is the observed amplitude of changes in optical density.

Provided that a comparison is made between curve $T_{t}\left(T\right)$ and curve S(T), the latter should also be transformed using a similar formula


(2)
$$s_{t}\left(T\right)=\left(1-\frac{S\left(T\right)}{S_{0}}\right)\cdot 100\%$$


Examples of graphs that compare curves obtained from the literature data $T_{t}\left(T\right)s_{t}\left(T\right)$ [138], are presented in Figure 4a,b. Figure 4a refers to curves 1 in Figure 2 and Figure 3, and Figure 4b refers to curves 4 in the same figures.

It can be seen that the topologically obtained curves are quite close to each other, and in the case of Figure 4b, they are close to each other even quantitatively.

Further, the dependence $C_{p}(S)$ will be a direct analogue of the phase portrait, but constructed in relation to the function S(T), since according to the construction of the function S(T), the function $C_{p}(S)$ is its derivative.

Accordingly, a construction based on the dependence $S(C_{p})$ can indeed be interpreted as an inverse phase portrait. Moreover, due to the topological similarity of the curves presented in Figure 4, direct phase portraits obtained on the basis of dependencies D(T) or $T_{t}\left(T\right)$, it is advisable to compare precisely with inverse phase portraits obtained on the basis of experimentally obtained dependencies $C_{p}(S)$.

It is significant that when constructing a direct phase portrait using the method [140,141], the calculation of the derivative is used (which in numerical calculations often leads to significant errors), while when constructing an inverse phase portrait, numerical integration is used, which gives more accurate results even when using the simplest method of rectangles.

#### 2.1.2. Comparison of Direct and Reverse Phase Portraits

This section uses the results of [10]. The goal is to show that thermograms provide significantly more complete information about the nature of phase transitions in solutions based on thermosensitive polymers than measurements carried out by turbidimetry. More precisely, they provide more visual information about the processes under consideration. Under certain conditions, similar information can be obtained from measurements carried out by turbidimetry methods, but to obtain it, an appropriate methodology is required. Its basis is the method of phase portraits.

We emphasize that a fairly large array of data reflecting the behavior of thermosensitive polymers and systems based on them was previously obtained using the turbidimetry method, which, in particular, is reflected in the literature [142,143,144,145,146]. In fact, this array reflects the results of research conducted by various scientific schools in this direction for more than twenty years. It is to involve these data in constructing the overall picture that it seems possible to develop methods for processing experimental data that make it possible to extract the most complete information from previously obtained data.

Figure 5, Figure 6, Figure 7 and Figure 8 show a comparison of phase portraits obtained using the curves presented in Figure 1 and Figure 3.

Figure 5 refers to curves 1 in the indicated figures, Figure 6 refers to curves 2, Figure 7 refers to curves 3, and Figure 8 refers to curves 4.

It can be seen that, as for previously studied systems [141,142,143,145], both phase portraits with acceptable accuracy fall into separate fragments, each of which is approximated by either a parabolic or linear dependence. Moreover, taking into account the accuracy factor, the direct and reverse phase portraits presented in these figures are characterized by a very specific topological symmetry. It manifests itself most clearly in Figure 7 and Figure 8: up to similarity transformation (axis scaling), these portraits are close to mirror-symmetrical.

This already creates the prerequisites for the development of appropriate classification characteristics, which are discussed below, although it should be noted that reverse phase portraits actually correspond to much higher accuracy than direct ones. This is due to the fact that the operation of numerical differentiation used in constructing phase portraits produces significantly larger errors than the operation of numerical integration.

Next, Figure 9a,b shows the curves obtained in one study [139] by the turbidimetry method (note that in addition to the cited works, the data method was also used in many other studies, in particular [146,147,148,149,150], and adequate comparison with the results obtained using other experimental methods was not always carried out).

Figure 10 shows phase portraits corresponding to these experimental curves.

It can be seen that the resulting phase portraits fall into two separate segments. One of them is approximated with acceptable accuracy by a parabolic dependence, and the second by a linear one. This means that the phase transition studied in a previous study [139] occurs in two stages, and the presence of the second stage can only be established by the method of phase portraits.

### 2.2. Discussion

Thus, the staged nature of phase transitions in solutions of thermosensitive polymers (and, consequently, gels based on them) is no exception to the rule. As shown in Figure 10, a two-stage phase transition turns out to be inherent even in such well-studied systems as the copolymer of NIPAAm with 2-HEA. Thus, we can conclude that the system studied in this work [139] is qualitatively different from the systems studied, in particular, in the aforementioned studies [140,151]. The systems studied in these works correspond to strictly parabolic phase portraits, which allows us to conclude that the phase transition in them occurs in one stage.

However, direct analysis of the dependence of the optical density of a solution on temperature does not always reveal the existence of different stages. We can speak with confidence about the existence of the second stage of the phase transition (Figure 10), partly due to the fact that a similar conclusion can be made based on a comparison of the results obtained by turbidimetry and differential calorimetry.

In particular, this means that it is permissible to raise the question of the resolution of various methods used to study phase transitions in solutions of thermosensitive polymers (the higher the resolution, the more accurately the various stages of the phase transition can be identified). The results presented above clearly show that the resolution of differential calorimetry is noticeably higher than the resolution of turbidimetry. Let us emphasize once again that the presence of the second stage of phase transition in the system studied in one study [139] went unnoticed by the authors of the cited work. Similarly, the presence of several stages of phase transition is extremely difficult to identify directly based on the analysis of turbidimetric measurement data obtained in the literature [147,148,149,150]. This may not be surprising, since the appearance of the curves presented in Figure 9 does not indicate this. Identification of the second stage of the phase transition required the development of a specific method for processing experimental data based on the construction of phase portraits.

Based on the results of a few studies [144,145,146], we can assume that in the system studied in one study [139], the phase transition occurs in at least two stages. The nature of the occurrence of these stages can be interpreted as follows: As shown in [144,145,146], the same processes that are responsible for the compaction of a macromolecular coil (for example, hydrophobic interactions) can lead to the formation of hydrophobic interpolymer associates (HIAs) as well as hydrophobic-hydrophilic associates (HHAs). Such objects are unstable meshes that exist in a dynamic mode, i.e., connections in them are continuously destroyed and arise again. Consequently, as the amplitude of the forces ensuring the formation of insoluble products increases, the destruction of HIA or HGA must inevitably first take place; this is the first obligatory stage of the phase transition, preceding the actual formation of the insoluble product. We emphasize that this stage must take place regardless of how many stages correspond to the processes occurring within a separate macromolecular coil or a separate complex based on a macromolecule. These processes occur at subsequent stages of the phase transition, when the conditions for the formation of HIA or HGA no longer exist. Moreover, it can certainly be assumed that, depending on the chemical structure of a particular polymer, the processes associated with the compaction of a macromolecular coil that occur after the destruction of HIA or HGA do not necessarily occur in an abrupt manner.

Apparently, this is precisely the case that is realized in the system studied in a previously cited study [139]. In this case, the first stage (the destruction of the dynamic mesh) takes place first, proceeding in an abrupt manner. This stage corresponds to a parabolic fragment of the phase portrait. We emphasize that, as shown in [140,151], a transition close to a jump-like one corresponds precisely to parabolic phase portraits. The second stage, corresponding to the linear fragment of the phase portrait, corresponds to relatively smooth processes leading to compaction of the macromolecular coil and, accordingly, to the formation of an insoluble product.

The main conclusion that follows from the comparison presented above (Figure 5, Figure 6, Figure 7 and Figure 8), as well as from Figure 10, is as subsequently described.

There is a real opportunity to classify systems based on thermosensitive polymers that undergo phase transitions according to the number of stages that characterize such a transition. The number of these stages can be determined, for example, by the method of inverse phase portraits using the differential scanning calorimetry method. The number of stages can be determined by the number of fragments (linear and parabolic) into which the phase portrait is divided. In principle, the number of such stages can also be estimated using the turbidimetry method; however, it has significantly lower accuracy (resolution) since it is necessary to use a numerical differentiation procedure, which introduces noticeable errors in the construction of the phase portrait. However, for qualitative analysis (that is, to identify qualitative classification features), in principle, it is possible to use phase portraits obtained with relatively low accuracy.

We emphasize that the use of phase portraits obtained with low accuracy based on available literature data is also of undoubted interest, since this makes it possible to use the richest experimental material accumulated in the literature over the past decades.

In particular, based on the analysis of the phase portraits presented in Figure 10 and their comparison with the phase portraits obtained in the literature [140,141,142,143,145,151], it is possible to draw certain qualitative conclusions; in any case, it can be judged that this particular phase portrait deviates from the parabolic one, and therefore, it is permissible to conclude that the corresponding phase transition occurs in more than one stage. For the use of qualitative criteria, this seems quite sufficient, especially since image recognition methods using neural networks are currently well developed [152,153]. In the future, these methods can be applied to the processing of such phase portraits as those shown in Figure 10. As is known, existing image processing methods make it possible to restore an image, including one containing significant defects [152,153].

Note, however, that the resolution of the proposed method depends on several factors and not just on the accuracy of constructing or reconstructing the phase portrait. The possibility of confidently identifying several stages of a phase portrait is determined, among other things, by how far the stages of the phase transition under consideration are separated from each other in temperature. If the temperatures of different stages are close, but the corresponding segments of the phase portrait overlap each other, it is quite difficult to separate them. However, since different stages of the phase portrait correspond to different physicochemical mechanisms, as the materials of the study show, the proposed method is applicable to a fairly wide range of temperature-sensitive systems.

We also note that the proposed method also makes it possible to find dependencies that describe phase transitions in more complex cases than those corresponding to Equation (8). For example, we use the data presented in curve 3 in Figure 9b. In Figure 11, these data are represented by empty circles (curve 1).

The phase portrait of this curve contains a section that is satisfactorily approximated by a linear dependence. This fragment of the phase portrait corresponds to the exponential dependence of the original curve.
(3)$$D=D_{0}\left(1-\exp-\frac{T-T_{0}}{\tau }\right)$$

Indeed, the derivative of function (3) is
(4)$$\frac{dD}{dT}=\frac{D_{0}}{\tau }\exp-\frac{T-T_{0}}{\tau }$$

Consequently, quantities (3) and (4), as one would expect, are linearly related. That is, the linear phase portrait corresponds to the case when the system under consideration is described by a first-order linear differential equation.

Figure 11 shows curve 2, corresponding to dependence of the form (3) with the following parameter values: $T_{0}=22\;K$, τ=10 K, $D_{0}=1.63$.

It satisfactorily approximates that portion of the experimental dependence that corresponds to the linear fragment of the phase portrait. Full circles in Figure 11 (curve 3) show values representing the ratio of experimentally measured values to values calculated using an approximation of the form (3). Curve 4 is plotted according to Formula (8) with the following parameter values: $T_{0}=34.8\;K$, τ=1.45 K, $D_{0}=1$. This curve approximates the given experimental data with satisfactory accuracy.

Therefore, we can conclude that the experimental results presented in Figure 11 must be satisfactorily described by a dependence of the form.
(5)$$D=D_{0}\frac{1-\exp-\frac{T-T_{01}}{\tau_{1}}}{1+\exp-\frac{T-T_{02}}{\tau_{2}}}$$
where indices 1 and 2 number the parameters related to two different sections of the phase portrait.

Curve 5 is constructed according to Formula (5). It can be seen that it describes the experimental data with satisfactory accuracy.

Formula (5) was obtained from the following considerations: a fragment of the phase portrait corresponding to the completion of the phase transition is approximated by a straight line. In accordance with Formulas (3) and (4), this means that the corresponding section of the experimental curve should be approximated with satisfactory accuracy by an exponential dependence, as demonstrated by a comparison of curves 1 and 2 in Figure 11. If we proceed from the assumption that the phase transition described by curve 1 occurs in two stages, then we should come to the conclusion that this exponential dependence corresponds to a stage occurring at a higher temperature (compared to the first). It can also be assumed that in this temperature range, the first stage is completed. Therefore, it is permissible to try to separate the stages by isolating the second. To do this, you can divide the values corresponding to the experimental data by the values calculated using exponential approximation (curve 2, Figure 11). The result is curve 3, which can be approximated by Formula (8). This approximation actually provides acceptable accuracy (curve 4, Figure 11). As a result, the experimental dependence can be approximated by the product of two curves, each of which describes one of the stages of the phase transition. This is what Formula (5) expresses. Its numerator corresponds to the curve approximating the second (final) stage of the phase transition, and the denominator corresponds to the first. It is appropriate to note that this approach has its limits of applicability. Namely, in order for Formula (5) to be adequate, the characteristic temperatures corresponding to the first and second stages of the phase transition must differ quite noticeably. The criterion in this regard is the nature of the phase portrait. If it is possible to clearly identify fragments corresponding to parabolic and linear dependencies, then Formula (5) is certainly applicable.

In Figure 12 shows a model curve, also constructed according to Formula (5). The phase portrait of this curve is shown in Figure 13.

By the nature of the curves, this portrait coincides with the phase portraits presented in Figure 10. Namely, it contains fragments approximated by a parabolic and linear dependence (as well as a transition region).

The presented example shows that, starting from the phase portrait, it is indeed possible to reconstruct quite complex dependencies that describe the phase transition. In addition, this example serves as additional confirmation of the adequacy of the proposed method. Namely, from Figure 13, it is clear that although the very nature of the curve under consideration is quite complex, nevertheless, the phase portrait shows areas that correspond to much simpler dependencies, including the one that corresponds to Formula (8).

The appearance of curve 1, Figure 11, and similar ones may be due to the following phenomena occurring during phase transitions. As shown in previous studies [144,145], the nature of the phase transition can be determined by the formation of supramolecular structures (hydrophilic interpolymer associates or hydrophobic-hydrophilic associates). They are formed due to the formation of unstable bonds between macromolecular coils. Therefore, the first stage of the phase transition (subject to the formation of such associates) will be associated precisely with the destruction of such structures. In this case, the decrease in the size of the macromolecular coil itself, caused by an increase in temperature, does not necessarily have to occur in an abrupt manner. The abrupt transition may be caused precisely by the destruction of associates of the above nature. Consequently, gradual compression of the macromolecular coil (which is expressed, among other things, in a change in the turbidity of the solution) can occur even after the associate is destroyed. Curve 1, Figure 11, its analogues, as well as Formula (5), presumably correspond to precisely this situation. The compression of the macromolecular coil itself proceeds quite smoothly, but at first it causes the destruction of the associate, which corresponds to the parabolic portion of the phase portrait. After the associate is destroyed, the compression of the macromolecular coil continues, but it continues to be quite smooth, which allows the use of exponential approximation (this corresponds to a linear fragment of the phase portrait).

This, however, does not exclude the need to develop methods that would make it possible to construct phase portraits using turbidimetry methods with higher accuracy than the numerical differentiation procedure allows. Indeed, as shown in Figure 4, Figure 5, Figure 6, Figure 7 and Figure 8, a comparison of the forward and reverse phase portraits obtained on the basis of turbidimetric and calorimetric measurements, respectively, is very informative. Such a comparison, among other things, makes it possible to most reliably verify the presence of several stages inherent in a specific phase transition.

In the future, such methods can be developed on the basis of radio engineering differentiation of the temperature dependence of the solution (or gel) transmission function. Note that similar methods have been used in plasma physics for more than a long time [154,155].

In relation to the case under consideration, the measurement scheme that allows one to directly obtain the derivative of the measured parameter with respect to temperature can be organized as follows: The temperature of the medium changes according to a law that corresponds to the sum of a linear (or function close to it) and harmonic oscillation. The optical vibration recording unit is configured so that it selects a component corresponding to a linear (or close to it) component and a separate component corresponding to a harmonic vibration. In this case, the amplitude of the harmonic oscillation is selected sufficiently small, i.e., such that the approximate equality is satisfied with acceptable accuracy.


(6)
$$T_{t}\left(T\left(t\right)+∆T\cos\omega t\;\right)\approx T_{t}\left(T\left(t\right)\right)+\frac{dT_{t}}{dT}∆T\cos\omega t$$


This relationship, in particular, shows that by isolating a harmonic at a ω frequency using radio engineering means, it is possible to determine the value of the derivative $\frac{dT_{t}}{dT}$ without using the numerical differentiation procedure. The value of this derivative is proportional, as Formula (3) shows, to the amplitude of the signal recorded at a given frequency.

Of course, the systems under consideration have very specific specifics, which does not allow the use of methods developed in plasma physics directly. Indeed, in previous [156] it was shown that phase transitions in systems of the type under consideration can be accompanied by hysteresis phenomena. There is also a factor of their pronounced inertia, etc.

However, even taking into account the above complicating factors, the problem under consideration for the current level of development of electronics is quite simple. For example, hysteresis phenomena can be taken into account by radio engineering isolating the second harmonic in the expansion of the function on the left side of Formula (3) in the series Taylor by parameter. Moreover, it is possible to significantly simplify its solution by using equipment interfaced with the user’s smartphone via a radio channel. This φ [157,158,159] has several other purposes, including the creation of new equipment for viscometric measurements. In the cited works, it was shown that the computing capabilities of modern smartphones make it possible to significantly simplify the electronic circuits of measuring equipment, i.e., the task of creating methods for direct radio engineering differentiation of the dependence of the optical density of a solution on temperature is indeed solvable. This, in the future, will significantly increase the resolution of turbidimetry in terms of determining the number of phase transition stages and using fairly simple methods.

## 3. Conclusions

There is no doubt that the more than significant amount of experimental data accumulated over the past decades in the field of studying phase transitions in solutions and gels based on thermosensitive polymers requires systematization.

Such systematization can only be built on the basis of one or another classification, which, in turn, presupposes the formulation of certain classification characteristics.

The study shows that, due to the heterogeneity of the available experimental material, at this stage of research, it is advisable to use a qualitative classification criterion, which is proposed to be the number of phase transition stages.

The study shows that this number can be established using the phase portrait method, and it is possible to use both direct and reverse phase portraits. In the latter case, the experimental curve is considered a derivative of the quantity under study with respect to the control parameter, and its numerical integration is used to construct the inverse phase portrait. The construction of reverse phase portraits is proposed in this work for the first time, and their advantage is the increased (compared to the method of direct phase portraits) accuracy of experimental data processing. It is also shown that, in relation to the study of systems based on thermosensitive polymers, it is advisable to apply the direct phase portrait method to the results of turbidimetric measurements and the reverse phase portrait method to calorimetric measurements. The phase portrait method also creates the basis for obtaining approximation curves that describe experimental data.

## 4. Materials and Methods

This study used experimental data presented in the literature [138,139].

Data processing is carried out using the phase portrait method, described, in particular, in the literature [140,141,151,160,161], as well as using the method of inverse phase portraits, proposed for the first time in this work.

The phase portrait method [140,141,151,160] is described below.

There is an experimentally obtained dependence of the measured parameter (for example, the turbidity of a polymer solution D) on the control parameter (for example, solution temperature T). Based on these data, the derivative of the measured parameter with respect to the control parameter is calculated, for example $\frac{dD}{dT}$, and then the dependence $\frac{dD}{dT}\left(D\right)$ is constructed.

There are known cases (for example, [140,151]), when such a dependence is a parabola:
(7)$$\frac{dD}{dT}=a_{2}D^{2}+a_{1}D+a_{0}$$
where $a_{i}$ are the coefficients obtained based on experimental data.

Equation (7) admits an analytical solution, which has the form
(8)$$D=D_{0}\frac{\exp\left[(T-T_{0})/\tau \right]}{1+\exp\left[(T-T_{0})/\tau \right]}$$
where $D_{0}$ is the amplitude of changes in optical density, $T_{0}$ is the temperature of the phase transition, τ is a constant characterizing the steepness of the phase transition, and which has the dimension of temperature.

In the literature [140,151], it was shown that this dependence well describes the experimental data corresponding to the case of a parabolic phase portrait.

Along with the method discussed above [140,141,151,160], which can be called the method of direct phase portraits, it is also permissible to use the method of inverse phase portraits when the experimentally measured quantity (for example, $C_{p}\left(T\right)$ [138]) is considered a derivative of a quantity obtained by integrating it over the control parameter. For example, it is permissible to consider the function
(9)$$S\left(T\right)=\int_{T_{m}}^{T}C_{p}dT$$
and build its dependence on the function $C_{p}(T)$.

As will be clear from what follows, this method complements the method of direct phase portraits, making it possible to more accurately identify the stages of phase transition in systems of the type under consideration.

The rationale for using direct and reverse phase portrait methods for the purposes of this work is as follows.

The advantage of the phase portrait method is its ability to identify the staged nature of phase transitions as well as to obtain differential equations that describe individual stages of a phase transition directly based on experimental data.

Namely, in the literature [141,142,143,144,145], it has been shown that the number of such stages can be different for different systems. Moreover, in other studies [143,144,145,162,163], it was shown that the staged nature of phase transitions can also be associated with fundamentally different processes. For example, for systems such as those studied in one study [144], it is determined by the formation of hydrophobic-hydrophilic associates (HHA), and for those studied in [145], it is determined by the formation of hydrophilic interpolymer associates (HIA). Such associates (both GGA and GIA) are unstable polymer networks, the bonds in which are continuously destroyed and created again [146].

The ability of macromolecules to form associates significantly depends [146] on the same factor that determines the formation of classical interpolymer complexes (or causes the coil-globule transition), i.e., on the formation of bonds between individual functional groups of macromolecules (hydrogen bonds, hydrophobic interactions). If the probability of the formation of such bonds is quite low, then interactions between different macromolecules prevail in the solution (if they are molecules of the same type, then HGA is formed; if they are different, HIA is formed). When the probability of the formation of such bonds increases (for example, due to an increase in temperature), HIA or HGA are destroyed, and instead of them, products are formed that include a relatively small number of macromolecules, for example, classical IPA [144,145,146].

The example of systems in which HGA and GIA are formed shows that a phase transition can indeed have a staged character, with different stages corresponding to the occurrence of processes of different natures.

The various stages of the phase transition in the cited works were identified using the phase portrait method discussed above. Namely, the topology of the phase portraits obtained in the literature [141,142,143,144,145] is different. In particular, phase portraits corresponding to the phase transition studied in [140,151] are strictly parabolic, which allows us to conclude that in this case the phase transition occurs in one stage. Phase portraits obtained in a previous study [145] break down into a set of linear sections. There are known cases when the phase portrait is a collection of parabolic sections [141,142].

This work shows that classification criteria that make it possible in the future to systematize a significant amount of data reflecting the features of phase transitions in systems based on thermosensitive (or, more broadly, stimulus-sensitive) polymers can be developed based on the phase portrait method. It is shown that, along with the method of obtaining direct phase portraits, it is also advisable to use the method of inverse phase portraits, which is proposed in this work for the first time.

## Figures and Tables

**Figure 1 gels-10-00395-f001:**
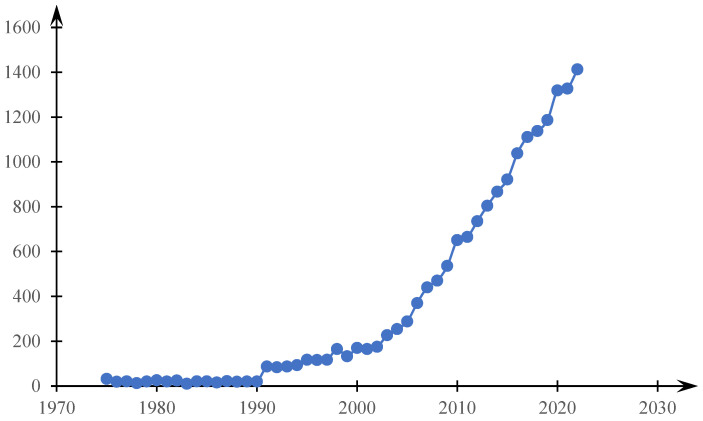
Statistics of search results for the keywords “thermosensitive”, “thermoresponsive polymer” in the WoS database by year.

**Figure 2 gels-10-00395-f002:**
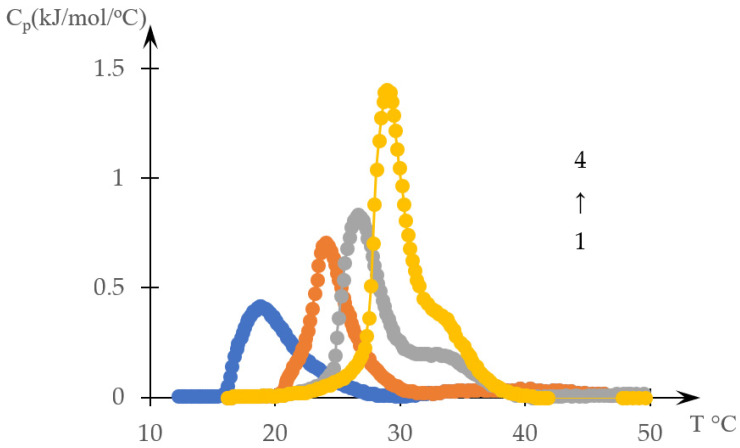
Thermograms of an aqueous solution of C12-PN-AzPy of varying molar masses (concentration: 0.5 mg/mL for polymers of molar masses 5K (1, blue) and 7K (2, orange); 1.0 mg/mL for polymers of molar masses 12K (3, gray) and 20K (4, yellow)) [138].

**Figure 3 gels-10-00395-f003:**
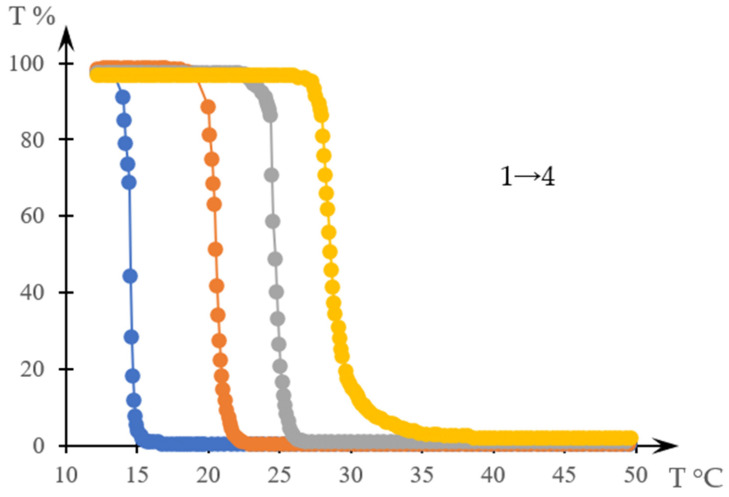
Turbidity curves of an aqueous solution of C12-PN-AzPy of varying molar masses (concentration: 0.5 mg/mL for polymers of molar masses 5K (1, blue) and 7K (2, orange); 1.0 mg/mL for polymers of molar masses 12K (3, gray) and 20K (4, yellow)) [138].

**Figure 4 gels-10-00395-f004:**
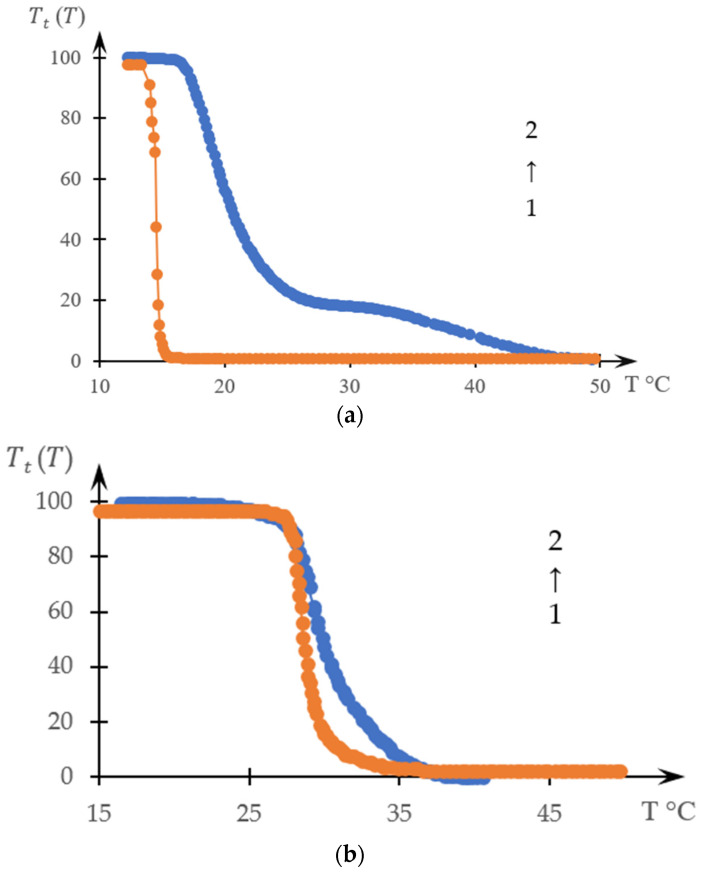
Comparison of curves $T_{t}\left(T\right)$ (1, orange) and $s_{t}\left(T\right)$ (2, blue) obtained from the data of [139]; (**a**) for curves 1 in Figure 2 and Figure 3, (**b**) for curves 4 in these figures.

**Figure 5 gels-10-00395-f005:**
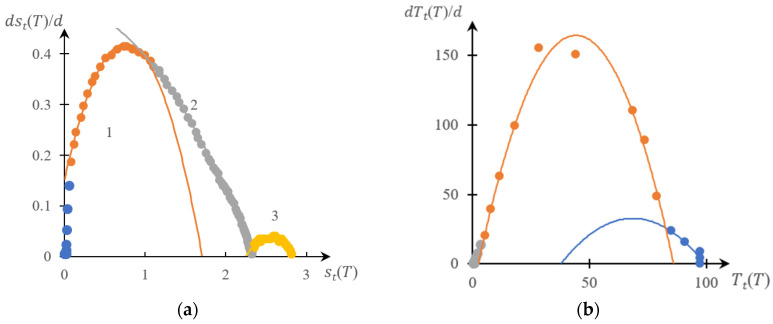
Reverse (**a**) and direct (**b**) phase portraits for curves 1 in Figure 2 and Figure 3. 1 and 3—parabolic sections of the phase portrait, 2—intermediate; different colors are used to mark different areas of the same phase portrait.

**Figure 6 gels-10-00395-f006:**
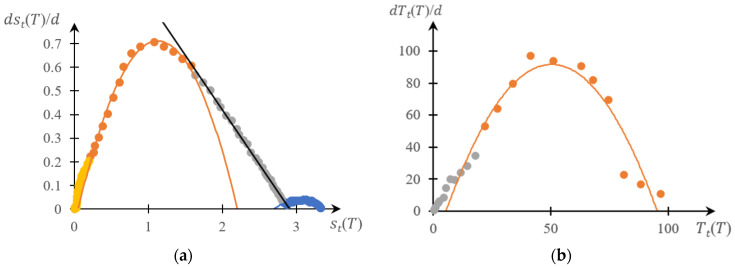
Reverse (**a**) and direct (**b**) phase portraits for curves 2 in Figure 2 and Figure 3; different colors are used to mark different areas of the same phase portrait.

**Figure 7 gels-10-00395-f007:**
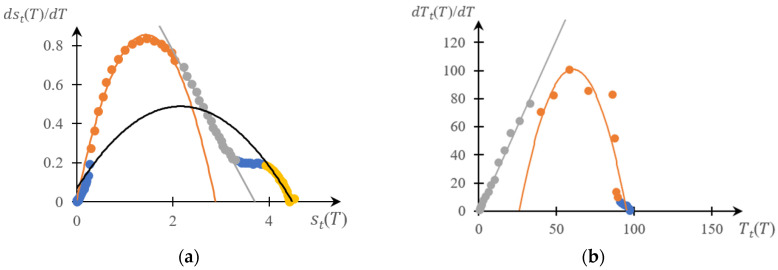
Reverse (**a**) and forward (**b**) phase portraits for curves 3 in Figure 2 and Figure 3; different colors are used to mark different areas of the same phase portrait.

**Figure 8 gels-10-00395-f008:**
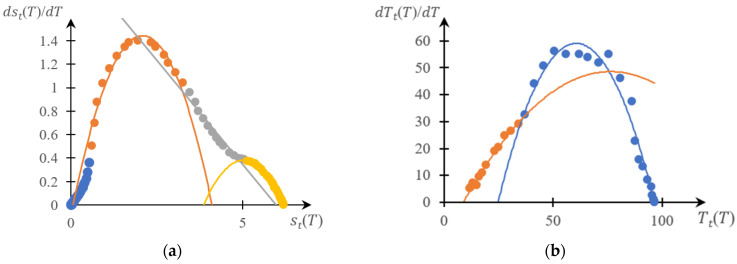
Reverse (**a**) and direct (**b**) phase portraits for curves 4 in Figure 2 and Figure 3; different colors are used to mark different areas of the same phase portrait.

**Figure 9 gels-10-00395-f009:**
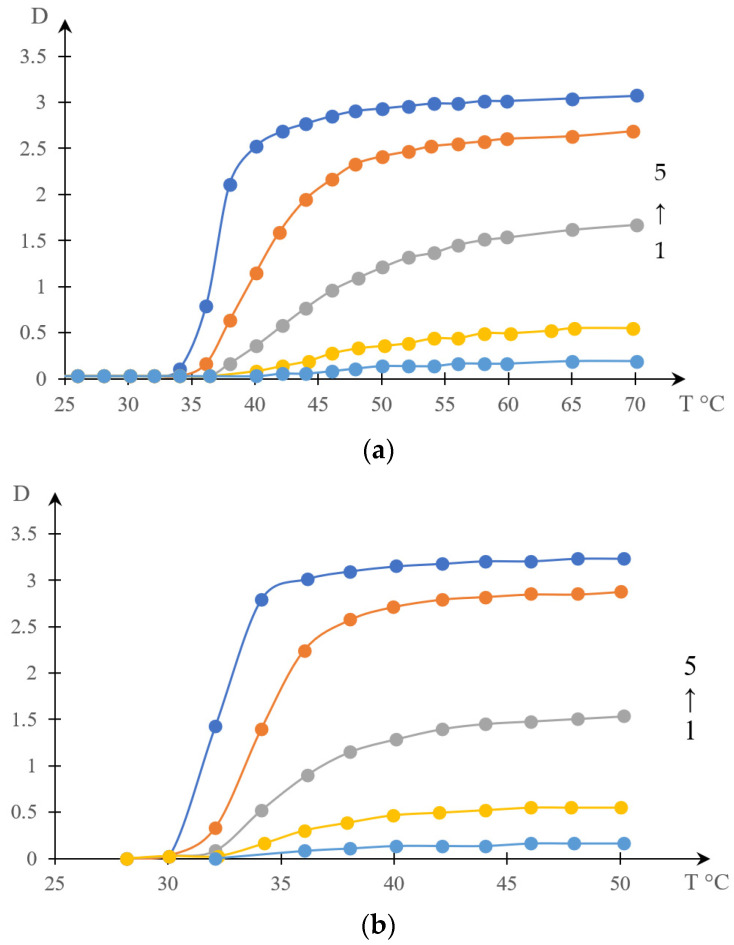
Dependences of the optical density of solutions of copolymer NIPAAm with 2-HEA with various ratios of hydrophilic chains 50:50 (**a**), 70:30 (**b**) on temperature: 0.0125% (1, light blue); 0.025% (2, yellow); 0.05% (3, gray); 0.1% (4, orange); 0.2% (5, dark blue) [139].

**Figure 10 gels-10-00395-f010:**
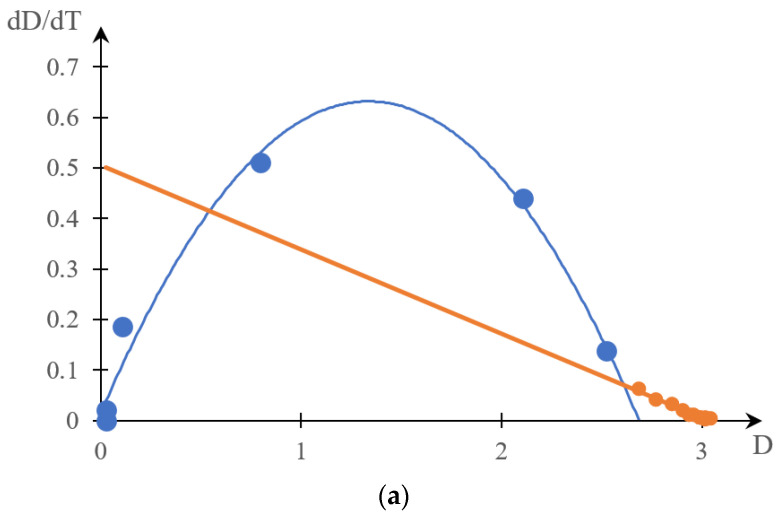
Phase portraits: curve 1 in Figure 9a (**a**), curve 2 in Figure 9a (**b**), curve 3 in Figure 9a (**c**), curve 1 in Figure 9b (**d**), and curve 1 in Figure 9b (**e**); blue color is a parabolic section, orange is a linear section.

**Figure 11 gels-10-00395-f011:**
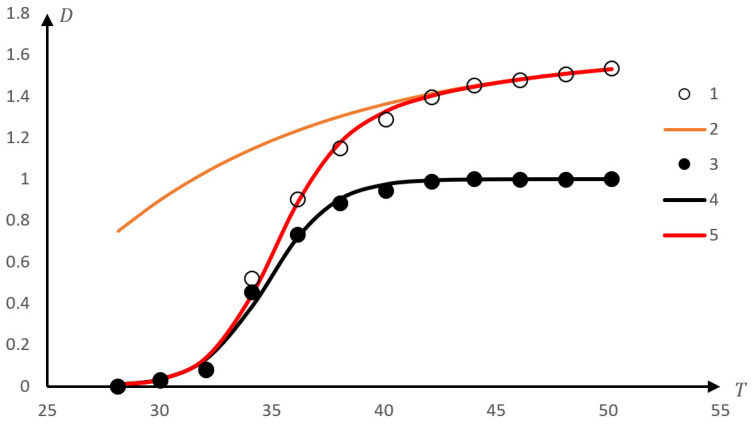
An example of processing experimental data, leading to a dependence of the form (5); curve 1—experimental data; curve 2 (orange)—approximation of the section of the experimental curve corresponding to the linear phase portrait using exponential dependence (6); curve 3—the result of dividing the values corresponding to the experimental data by the result of approximation according to Formula (6); curve 4 (black)—approximation of the curve 3 using a formula of the form (2); curve 5 (red)—approximation of the original experimental data using a dependence of the form (8).

**Figure 12 gels-10-00395-f012:**
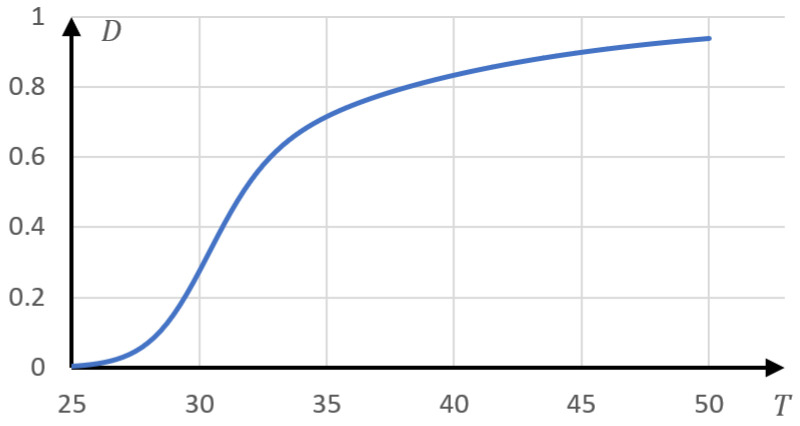
Model curve constructed according to Formula (5); $T_{01}=22\;K$, $\tau_{1}=10\;K$, $T_{02}=30\;K$, τ=1.2 K, $D_{0}=1$.

**Figure 13 gels-10-00395-f013:**
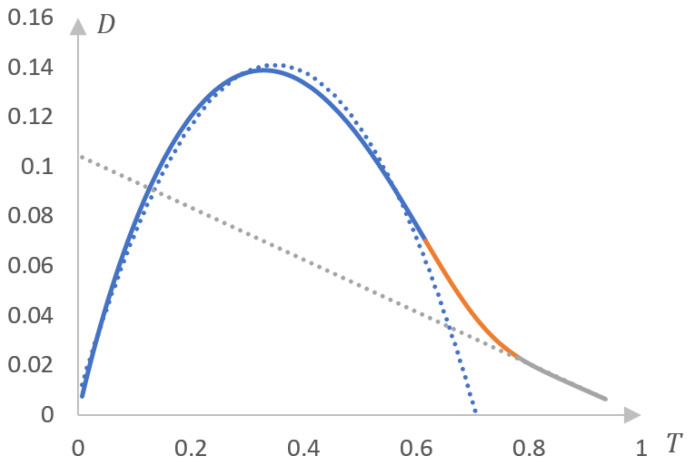
Phase portrait of the model curve, Figure 12; blue—parabolic section of the phase portrait, gray—linear, orange—intermediate; parabolic and linear approximations are shown by dotted lines in the corresponding colors.

## Data Availability

The original contributions presented in the study are included in the article/Appendix A, further inquiries can be directed to the corresponding authors.

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
