# Peer review of "The Method of Direct and Reverse Phase Portraits as a Tool for Systematizing the Results of Studies of Phase Transitions in Solutions of Thermosensitive Polymers"

_gels, 2024, doi:10.3390/gels10060395_

Round 1

Reviewer 1 Report

Comments and Suggestions for Authors

The paper titled “The method of direct and reverse phase portraits as a tool for systematizing the results of studies of phase transitions in solutions of thermosensitive polymers” by A. Bakirov, E. Kopishev, K. Kadyrzhan, E. Donbaeva, A. Zhaxybayeva, M. Duisembiyev, F. Suyundikova, and I. Suleimenov shows how to construct direct and inverse phase portraits together with analyses of data obtained from several literature examples.

The Introduction contains 137 references and feels more like a review paper. This part should be significantly shortened since it is not important for the core subject of the paper.

Overall, the paper is extremely lengthy. The core method introduced in this paper is the construction of inverse phase portraits, which can be described a bit more concisely.

The only novelty is equation (3) and its normalization in equation (5) to use the phase portraits method (which was already published in Ref. 140-144) to identify the possible number of stages of the phase separation process in polymeric systems.

I agree with the authors that direct and inverse phase portraits can be helpful in qualitative classification of the number of stages of phase separation. This information can be subsequently used to assess experimental data more carefully (for example, for better decomposition of the whole phase separation profile into components).

However, the phase portrait method does not uncover much quantitative information. The whole method is solely based on the premise that the phase transition has the shape of a symmetric sigmoidal function (equation (2)) and the differential equation derived from it, which in the simplest case (i.e. one-stage phase separation) has a form of parabola. As soon as the phase separation is more complex (i.e., two-stage or sigmoidal but non-symmetric), the differential equation cannot be written in simple parabolic form and direct connection with experimental data in only informative as described above.

The paper is not structured well (lengthy Introduction and Discussion parts), and the novelty is small, therefore, I cannot recommend the paper for publication.

Comments on the Quality of English Language

English needs some fixing. There are many small mistakes/errors. The references contain many mistakes, errors and typos (e.g. wrong formatting of the title or names, missing DOI, etc.).

Reviewer 2 Report

Comments and Suggestions for Authors

The theme chosen for research by the authors, is daring and of interest given that thermosensitive polymers and their phase transformations are defining for their various applications in medicine, obtaining medicines, food, industry, etc. What should be remembered and emphasized would be:

1. What is the starting point of this broad analysis, what is the need for generalization and, as a follow-up, of the graphic/mathematical studies undertaken in the presented studies.

2. What is the predictive power of these models, or of these generalizations.

3. What are the limitations of the method?

4. The graphic representations must be standardized from the point of view of the resolution.

5. A careful rechecking of the list of bibliographic references is needed.

Comments on the Quality of English Language

Quality of English language should be improved by a carrefuly recheck in typewritting and spelling.

Reviewer 3 Report

Comments and Suggestions for Authors

The paper is nice and well written, but authors should stress more in the Abstract and Introduction what is the novelty of the paper, what was done for the first time.

Moreover, the conclusions are very general and only the last pragraph deals somehow with the work itself, the rest if current challanges and maybe future perspectives. I also advise to revise that part.

Round 2

Reviewer 1 Report

Comments and Suggestions for Authors

After the changes done by the authors, the manuscript can be published. However, I still have several comments.

In the new Figure 11, the curves are labelled unclearly, and it is hard to find the correspondence with text. Also, in the sentence “Curve 5 is constructed according to formula (8). It can be seen that it describes the experimental data with satisfactory accuracy.”, it is not clear which “Curve 5”?

New formula (8) comes out of blue. There is no justification for its form. Why is it the ratio of equations (2) and (6)? Moreover, the formula (8) has a singularity at T = T02. So, the useful form should have “+ exp” in the denominator and not “- exp”. But I guess this is just a typo.

I appreciate that the authors gave some explanation for the linear parts of the phase portrait.

Comments on the Quality of English Language

The English is fine.

Reviewer 2 Report

Comments and Suggestions for Authors

The manuscript has been sufficiently improved for a possible publication in Gels.

Comments on the Quality of English Language

Minor editing of English language is required
